# How Soda Ingestion Facilitates the Distinction between a Killian–Jamieson Diverticulum and a Malignant Thyroid Nodule

**DOI:** 10.3390/diagnostics13193128

**Published:** 2023-10-05

**Authors:** Tsung-Jung Liang, Shiuh-Inn Liu, Chia-Ling Chiang

**Affiliations:** 1Division of General Surgery, Department of Surgery, Kaohsiung Veterans General Hospital, Kaohsiung 813414, Taiwan; tjliangmd@gmail.com (T.-J.L.);; 2School of Medicine, National Yang Ming Chiao Tung University, Taipei 112304, Taiwan; 3Department of Radiology, Kaohsiung Veterans General Hospital, Kaohsiung 813414, Taiwan

**Keywords:** Killian–Jamieson diverticulum, thyroid nodule, thyroid cancer, microcalcification, ultrasonography, esophageal diverticulum, esophagography, ring-down artifact

## Abstract

A 66-year-old woman presented with an incidental left thyroid nodule during a health examination. She had no voice change, shortness of breath, cough, or dysphagia. Repeated sonography showed a dynamic change of the lesion, which was more evident following soda consumption. A subsequent esophagography confirmed the diagnosis of a Killian–Jamieson diverticulum. This rare left-sided pharyngoesophageal diverticulum is often asymptomatic. On a sonography, air bubbles in the esophageal lumen can cause a ring-down artifact that mimics microcalcifications, which are characteristic of thyroid malignancy, and misdiagnosis may lead to unnecessary interventions, including fine-needle aspiration or thyroidectomy. A dynamic ultrasound, specifically done during soda consumption, offered a simple diagnostic distinction. No surgical intervention was pursued; the patient was monitored in the clinic.

A 66-year-old woman presented to the surgical clinic for further management of an incidental left thyroid nodule detected during a routine health examination (Figure 1, arrows). She denied experiencing voice changes, dysphagia, cough, or shortness of breath. Initial assessments documented in the referral sheet indicated normal thyroid function tests, and the fine-needle aspiration cytology showed the presence of some benign-appearing follicular cells. Repeated sonography showed a 2.3 cm lesion on the posterior aspect of the left thyroid. Notably, dynamic changes in size and shape were observed during swallowing (Figure 2 and Figure 3, Appendix A). A subsequent esophagography confirmed the diagnosis of Killian–Jamieson diverticulum (Figure 4, arrows).

A Killian–Jamieson diverticulum is a rare, usually asymptomatic, pharyngoesophageal diverticulum on the left side [1]. Occasionally, it may lead to symptoms such as dysphagia, regurgitation, a sensation of a lump in the throat (globus sensation), and even aspiration pneumonia, as reported in some cases [2]. Anatomically, a Killian–Jamieson diverticulum protrudes from a gap in the anterolateral wall of the esophagus, located below the cricopharyngeal muscle [3]. This is in contrast to the more commonly encountered Zenker’s diverticulum, which originates in the posterior esophageal wall and is situated above the cricopharyngeal muscle [3]. Despite their differing locations, both conditions are classified as false diverticula since the pouch only contains mucosa and submucosa, lacking the full layers of the esophageal wall [3]. On a sonography, air bubbles in the esophageal lumen can cause a ring-down artifact that mimics microcalcifications, which are characteristic of thyroid malignancy, and misdiagnosis may lead to unnecessary interventions, including fine-needle aspiration or thyroidectomy [1,4,5]. To differentiate between a Killian–Jamieson diverticulum and thyroid nodules, the use of a dynamic sonography during swallowing plays a crucial role in identifying distinct changes in the contour of the lesion. Moreover, the administration of soda during the examination can enhance the passage of air through the diverticulum, intensifying the discernible changes and facilitating accurate differentiation (Appendix A) [6]. In this scenario, soda plays a role similar to that of a contrast medium, and its application aligns with the evolving field of contrast-enhanced ultrasounds [7]. In the case presented, the patient was referred to a thoracic surgeon for potential surgical intervention. However, she chose to pursue conservative management due to the absence of symptoms.

## Figures and Tables

**Figure 1 diagnostics-13-03128-f001:**
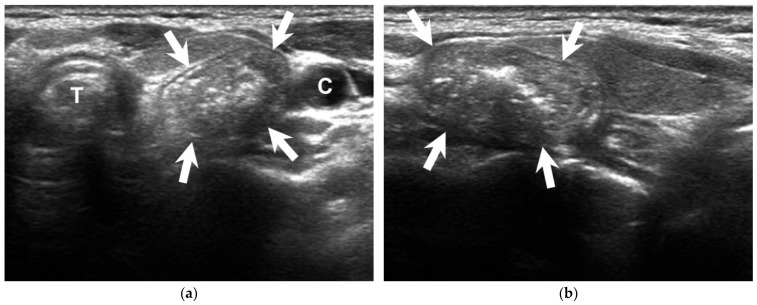
Transverse (**a**) and sagittal (**b**) sonographic images show a heterogeneous mass (arrows) with punctate echogenic foci on the posterior aspect of the left thyroid gland. C = carotid artery, T = trachea.

**Figure 2 diagnostics-13-03128-f002:**
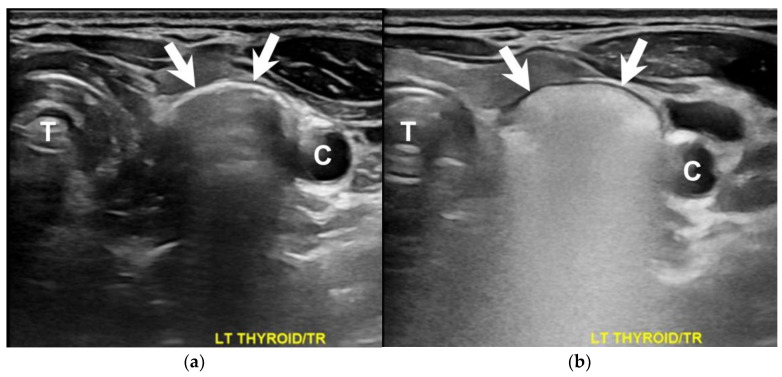
A transverse view of the dynamic sonography during swallowing depicted discernible changes in the lesion (arrows) prior to (**a**) and following (**b**) soda administration. C = carotid artery, T = trachea.

**Figure 3 diagnostics-13-03128-f003:**
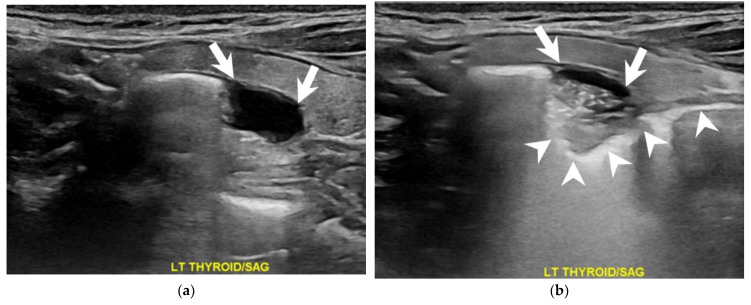
A sagittal view of the dynamic sonography during swallowing illustrated noticeable changes in the lesion (arrows) before (**a**) and after (**b**) the introduction of soda. The flow of soda traversed through the lesion and descended into the esophagus (arrowheads).

**Figure 4 diagnostics-13-03128-f004:**
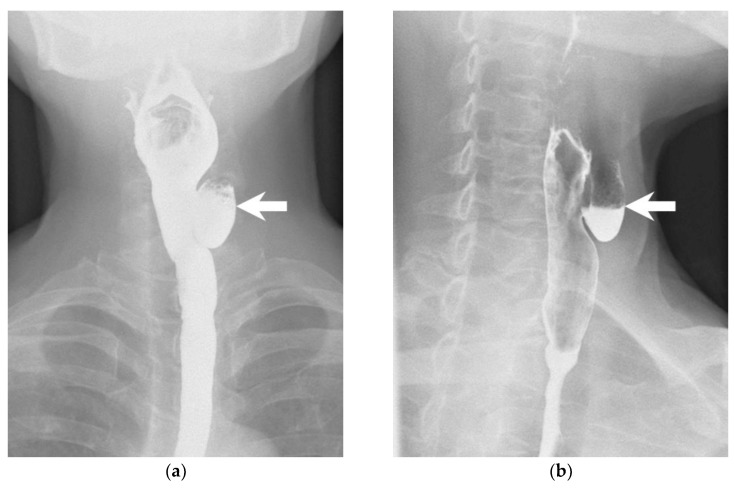
An esophagogram obtained with the patient in a frontal (**a**) and lateral (**b**) position shows a 1.8 cm Killian–Jamieson diverticulum on the left side (arrows).

## Data Availability

Data are contained within the main text of the manuscript.

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
