# Peer review of "How Soda Ingestion Facilitates the Distinction between a Killian–Jamieson Diverticulum and a Malignant Thyroid Nodule"

_diagnostics, 2023, doi:10.3390/diagnostics13193128_

Round 1
Reviewer 1 Report
Thank you for the opportunity to review your work.
I find it relevant for publication but do have a few comments I think would be relevant.
First of all, I think you neglect to include relevant references. Ultrasound has been used to assess the diverticula of the proximal esophagus before and your choice of references seems quite narrow. Secondly, I would advise you to point out the distinction between the Killan-jamieson diverticula and the Zenkers in your article.
Furthermore, I do believe you overstate the risk of confusing this pathology with a carcinoma of the thyroid for the sake of underscoring relevance. That is not needed.
Lastly, I think the novel element of your work is the use of soda as a contrast medium- I think a reference to the emerging field of contrast-enhanced ultrasound would be relevant.
Reviewer 2 Report
I think it is an interesting article.
Figures are ok.
It is useful reading this articles as it suggests a simple and practical method which can help detect and diagnose Killian diverticulum without the necessity to perform esophagogram.
References are ok.
Reviewer 3 Report
In this study, the authors present a very interesting case of Killian-Jamieson diverticulum which, like the more common Zenker diverticulum, is a false diverticulum of the esophagus representing an outpouching of mucosa through a muscular defect. Moreover, the authors highlight the possibility of misdiagnosing it with a thyroid nodule, leading to unnecessary interventions and complications.
Regarding the manuscript, it is concise and informative enough. In the supplementary videos, it could be very helpful to add an arrow pointing out the area of the diverticulum, which shows the dynamic changes in size and shape during swallowing. Very nice differential diagnosis based on the us findings. Moreover, the authors highlighted the helpful role of soda swallowing during dynamic sonography. Finally, it could be very interesting if the authors reported the complication accompanied by the diverticulum (such as rupture, or abscess), because patients with Killian-Jamieson diverticulum may present with complication and not asymptomatically as in this case.
In general, a very nice presentation of this educational interesting case.
Reviewer 4 Report
The authors present even interesting case of Killian-Jamieson diverticulum in 66-year lady. The case is well documented however some points should be discussed:
#minor:
It would be better write „arrows” than „arrow” as we have plural.
#major:
< Did the authors perform fine needle aspiration biopsy of this diverticulum?? And what is more interesting, they received the “normal result?” So, what does it mean “normal results” - “thyreocytes??”
< The authors say, that diagnostics which they preformed – ultrasonography, aesophagography, dynamic usg, soda’s swallowing etc. can help to avoid mistakes like biopsy or thyroidectomy, but they performed FNAB. So it should be discussed.
< The patient was not qualified to diverticulectomy? Did the authors propose it to the patient? If not, why?
English used quite correctly.
Round 2
Reviewer 4 Report
The authors addressed to the comments, so I do not have any more.
It can be.